# A deep learning-based system for automatic detection of emesis with high accuracy in *Suncus murinus*
Zengbing Lu[1,6], Yimeng Qiao[2,6], Xiaofei Huang[1], Dexuan Cui[1], Julia Y. H. Liu[1], Man Piu Ngan[1], Luping Liu[1], Zhixin Huang[3], Zi-Tong Li[1], Lingqing Yang[1], Aleena Khalid[1], Yingyi Deng[1], Sze Wa Chan[4], Longlong Tu[5] & John A. Rudd [1]✉

Quantifying emesis in *Suncus murinus* (*S. murinus*) has traditionally relied on direct observation or reviewing recorded behaviour, which are laborious, time-consuming processes that are susceptible to operator error. With rapid advancements in deep learning, automated animal behaviour quantification tools with high accuracy have emerged. In this study, we pioneere the use of both three-dimensional convolutional neural networks and self-attention mechanisms to develop the Automatic Emesis Detection (AED) tool for the quantification of emesis in *S. murinus*, achieving an overall accuracy of 98.92%. Specifically, we use motion-induced emesis videos as training datasets, with validation results demonstrating an accuracy of 99.42% for motion-induced emesis. In our model generalisation and application studies, we assess the AED tool using various emetics, including resiniferatoxin, nicotine, copper sulphate, naloxone, U46619, cyclophosphamide, exendin-4, and cisplatin. The prediction accuracies for these emetics are 97.10%, 100%, 100%, 97.10%, 98.97%, 96.93%, 98.91%, and 98.41%, respectively. In conclusion, employing deep learning-based automatic analysis improves efficiency and accuracy and mitigates human bias and errors. Our study provides valuable insights into the development of deep learning neural network models aimed at automating the analysis of various behaviours in *S. murinus*, with potential applications in preclinical research and drug development.

Emesis, the forceful expulsion of stomach contents through the mouth, is a common occurrence as a side effect of drugs and as a symptom of disease. The underlying mechanisms behind these responses likely evolved to protect vomiting-capable species against food poisoning[1]. Many studies in this research domain have been conducted to discover anti-emetic drugs against a variety of stimuli that activate multiple levels of the brain-gut axis (e.g., chemotherapy)[2,3]. Commonly used laboratory animals, including rats and mice, lack an emetic reflex and therefore, most studies have been conducted using specialised laboratory species, including ferrets, dogs, and cats[1]. *Suncus murinus* (*S. murinus*), commonly known as the house musk shrew, is an insectivore that has been used increasingly in emesis research since the late 1980s[4,5]. The quantification of emesis in *S. murinus* has primarily relied on direct observation and/or watching videos of recorded behaviour. Quantification of emetic episodes is commonly accomplished by manual counting, which is tedious and potentially time-consuming, with observation periods extending to 72 h[6].

In recent years, major progress has been made in the field of artificial intelligence, particularly with the development of deep learning methodologies. Furthermore, the rapid advancements in computing capabilities, particularly in graphics processing units (GPUs), along with the availability of new open-source deep learning libraries, such as PyTorch[7], Keras[8], and Tensorflow[9], have dramatically accelerated the widespread adoption of deep learning techniques. These advancements have permeated various scientific domains. For instance, convolutional neural networks (CNNs) have become

[1]Emesis Research Group, School of Biomedical Sciences, Faculty of Medicine, The Chinese University of Hong Kong, Shatin, New Territories, Hong Kong. [2]Department of Chemical and Biological Engineering, The Hong Kong University of Science and Technology, Clear Water Bay, Kowloon, Hong Kong. [3]School of Health Sciences, University of Manchester, Manchester, UK. [4]School of Health Sciences, Saint Francis University, Tseung Kwan O, New Territories, Hong Kong. [5]Department of Pediatrics, USDA/ARS Children's Nutrition Research Center, Baylor College of Medicine, Houston, TX, USA. [6]These authors contributed equally: Zengbing Lu, Yimeng Qiao. ✉e-mail: jar@cuhk.edu.hk

indispensable for tasks involving computer vision[10], while recurrent neural networks have been tailored for analysing temporal dynamics[11]. Recently, deep learning models have showcased their unique potential to improve automated animal behaviour detection. For example, DeepLabCut is a widely used toolkit for experimental animal pose estimation, enabling predictive analysis of animal behaviour with neural networks trained on annotated key point data[12]. Similarly, previous studies have demonstrated high accuracy in various rodent behaviour analysis tasks, encompassing limb motion analysis and the phenotyping of inflammatory nocifensive behaviour[13,14]. Social LEAP Estimates Animal Poses (SLEAP) tool has been developed for multi-animal pose tracking across diverse species, ranging from flies and bees to mice and gerbils[15]. Recently, Yurimoto et al. developed a Full Monitoring and Animal Identification system, enabling longitudinal tracking of the three-dimensional (3D) trajectories of each individual marmoset in a group of marmosets under free-moving conditions by integrating video tracking, light detection and ranging, and deep learning[16]. However, this methodology typically requires biologists to manually annotate the key points of animal skeletons in each video frame, a process that is tedious and unwieldy for experimental technicians. To address this challenge, numerous end-to-end models have arisen, providing enhanced user-friendliness. DeepEthogram, developed by Bohnslav et al. in 2021, offers a streamlined process for directly classifying various behaviours of mice and flies from unprocessed videos[17]. Geuther et al. used 3D CNN for grooming behaviour classification, and this tool can detect grooming behaviour with human observer-level performance[18]. Scratch-AID has exhibited competitive performance in identifying mouse scratching behaviour[19]. Nonetheless, for infrequent, short-duration behaviours, such as emesis in *S. murinus*, which pose greater challenges for automated detection, a specialised approach capable of achieving high sensitivity, specificity, and generalisability and replacing human observers remains to be developed.

In this investigation, our objective was to develop an innovative deep learning system with the ability to precisely quantify emetic behaviour in *S. murinus* automatically using data from motion-induced emesis experiments. The process commenced with video recording to capture emetic behaviours, followed by annotation of these recorded behaviours. Several challenges were used to induce emesis, including resiniferatoxin (RTX)[20], nicotine[21], copper sulphate[22], naloxone[23], U46619[24], cyclophosphamide[25], and exendin-4[26,27]. Subsequently, the annotated behaviours were subjected to model training using the R3D Model architecture and subsequent testing of the model. The applicability of the approach was also tested by examining the anti-emetic potential of palonosetron against the emesis induced by cisplatin[22,28]. Our aim was to generalise the model so it can be applied to the automatic detection of emesis induced by any challenge in *S. murinus*.

## Results

### The overall workflow
Our methodology for creating the Automatic Emesis Detection (AED) system to detect and measure emetic behaviour involved four primary stages (Fig. 1A): (1) capturing emetic behaviour triggered by provocative motion via video recording; (2) manually marking emesis frames in all recorded videos to generate training and testing datasets; (3) developing a deep neural network and training it with the annotated videos, followed by evaluating its effectiveness using validation videos; and (4) examining the adaptability of the trained neural network in emesis models induced by different stimuli and in anti-emetic drug screening studies.

### Experimental set-up for video recording
We developed a straightforward, simple video recording setup crucial for ensuring the stability of trained models and enabling adoption by other research teams. *S. murinus* behaviours were captured within a non-transparent Perspex box measuring $20 \times 18 \times 21$ cm to minimise external visual distractions (Fig. 1B). Illumination was provided from the top of the box, and the animals had unrestricted movement on bedding placed on the floor. The C930e camera captured behaviours from an overhead perspective at a rate of 30 fps (Fig. 1B). A typical instance of recording emetic behaviour

can be observed in Video 1. The camera's recording software (Logitech C930e Business Webcam driver and software) allowed for adjustments in magnification, resolution, and brightness, ensuring consistent video recording. In essence, this custom-designed recording enclosure facilitated high-quality video capture of emetic behaviours within a stable and reproducible setting.

### Manual annotation
We recorded 189 videos, comprising nine emesis conditions (123 videos for motion-induced emesis and 66 videos for eight drug-induced emesis conditions; Fig. 1C, D). To supervise the training and testing of the model, we manually annotated the videos. Given the brief duration of emesis in *S. murinus* (typically lasting only 1–2 s), we visually identified the start and end times of the emetic behaviour. Each frame within this interval was labelled as 'emesis', adhering to standard video annotation practices. Frames outside of these intervals were marked as 'non-emesis'.

To train and test the model in motion-induced studies, we converted the annotated videos into images and divided them into training and test datasets (at approximately a 9:1 ratio). It is worth noting that we exclusively used motion-induced videos for training and testing (Fig. 1C), reserving the recordings of emesis induced by the eight drugs for subsequent validation and generalisation analyses (Fig. 1D). The emesis episodes for individual animals in the training, testing, and generalisation studies are summarised in Fig. 1E, F, respectively.

### Model design and training
Emetic behaviours are considered to be rare occurrences under normal conditions in *S. murinus*. However, when challenged, they may exhibit multiple episodes of retching and/or vomiting (R + V), interspersed with non-emetic phases. The emetic episodes in this species are exceedingly brief (~1–2 s), and characterised by a series of retches followed by an expulsion event (vomiting; see Video 1). A distinctive feature of emesis is instantaneous twitching movements of the body. The continuous rise and fall of the animal's back can be used to recognise movement patterns associated with emetic behaviour. Temporally, these movements exhibit periodic rises and fall, while spatially, the relative positioning of the back and abdomen of *S. murinus* presents a relatively unique feature during such episodes.

Numerous deep learning models designed for action recognition aim to capture such spatiotemporal features. However, models such as R(2 + 1)D[29] and P3D[30] decompose spatiotemporal characteristics into separate parts for learning, which may not fully grasp comprehensive knowledge and global behaviour patterns. To better accommodate spatiotemporal information, we opted for an adapted R3D model[29] for learning purposes. The R3D model used in this study is a fully 3D ResNet model comprising 18 layers, supplemented with two additional transformer blocks. Fully 3D convolutions have demonstrated strong action recognition capabilities by incorporating temporal reasoning based on 2D still images. Furthermore, we incorporated additional transformer blocks between the initial residual blocks, leveraging an attention mechanism to establish global dependencies among the inputs, which addresses the limitation of small perspective fields in vanilla CNN filters. A schematic diagram of the R3D model is presented in Fig. 2.

One hundred and two videos, comprising 65,115 emesis frames (1447 emetic events) and 65,115 non-emesis frames, were employed for training. Throughout the training process, the loss, which represents the discrepancy between model predictions and manual annotations, decreased steadily (Fig. 3A), while the prediction accuracy, defined as the correct prediction of both emesis frames and non-emesis frames among all frames, increased (Fig. 3B). After the completion of five epochs (where one epoch entails model training on the complete training dataset once), the accuracy reached a plateau (Fig. 3A). The prediction accuracies consistently exceeded 97% for all input lengths, with a slight improvement observed with longer input lengths (Fig. 3B).

### Model evaluation on test datasets
We evaluated the effectiveness of the trained prediction models using 21 videos capturing motion-induced emesis. The heatmap shown in

**Fig. 1 | The overall workflow for setting up a video recording system for recording emetic behaviour in *Suncus murinus*. A** The workflow for developing the Automatic Emesis Detection (AED) tool for automatic detection and quantification of emetic behaviour. **B** Images showing the customised video recording system for emetic behaviour. **C** A schematic diagram showing the provocative motion-induced emesis model in *S. murinus*. **D** A schematic diagram showing the emesis model induced by different drugs, including resiniferatoxin (RTX), nicotine, copper sulphate, naloxone, U46619, cyclophosphamide, exendin-4 and cisplatin. **E** Episodes of retching (R) + vomiting (V) for individual animals with motion-induced emesis. **F** Episodes of R + V for individual animals in the emesis model induced by different drugs.

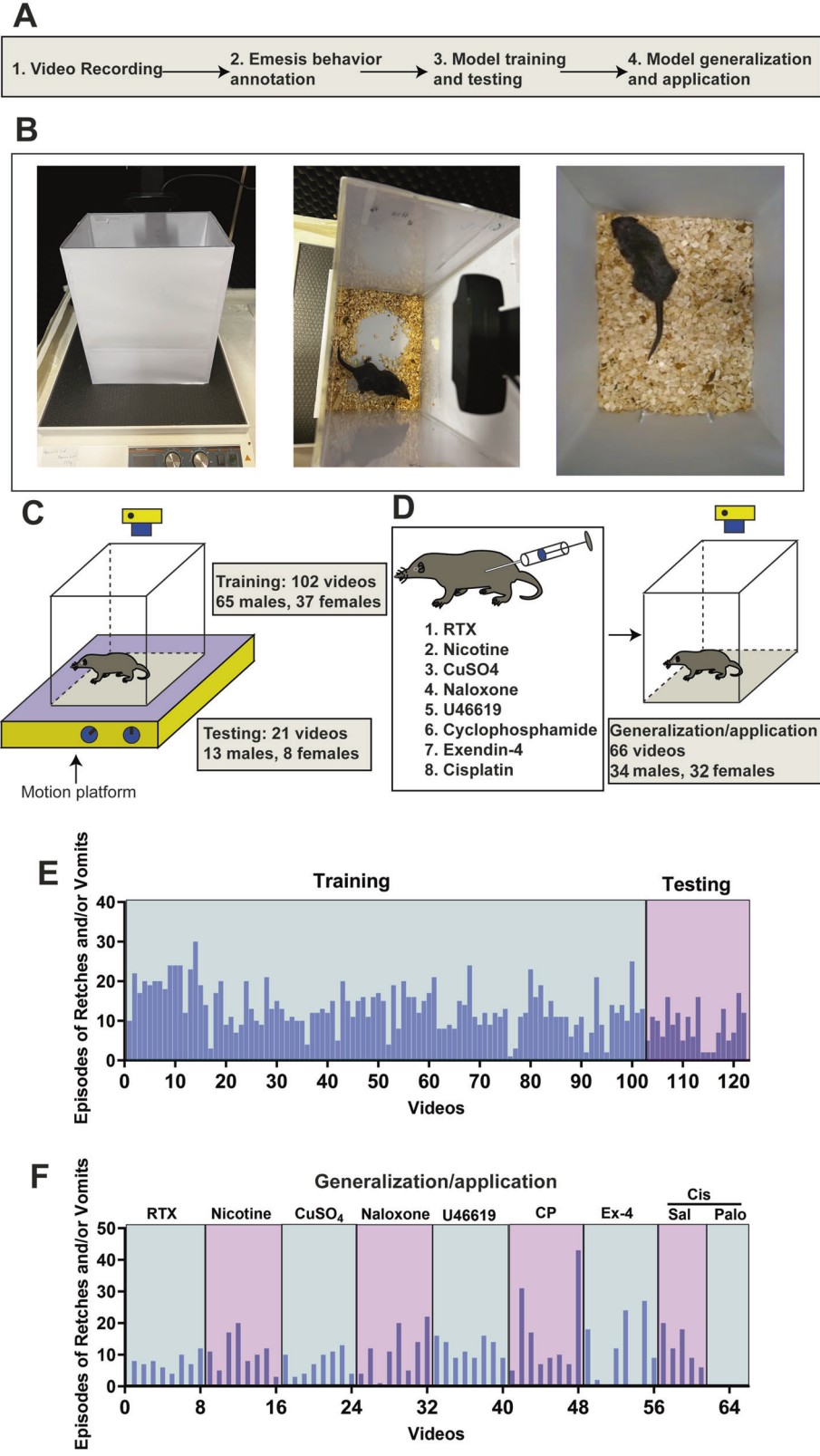

Fig. 4A illustrates the emetic profiles of individual animals exposed to provocative motion. For an emetic event, the trained model was unable to distinguish retching from vomiting, as they were too similar, thus an emetic event was defined as one episode of retching and/or vomiting. One hundred and ninety-two emetic events were analysed in these recordings. Notably, the model misclassified only one emetic event as non-emesis, resulting in a prediction accuracy of 99.42%. Moreover, the accuracy of detecting non-emesis frames in the testing dataset was 98.15%.

To further assess the performance of our model in motion-induced emesis, we employed standard metrics from deep learning methods, including precision, recall, and the F1 score. The confusion matrix of the test dataset is shown in Supplementary Table 1. The precision, recall, and F1 score in the evaluation dataset were 0.9145, 0.9942, and 0.9529,

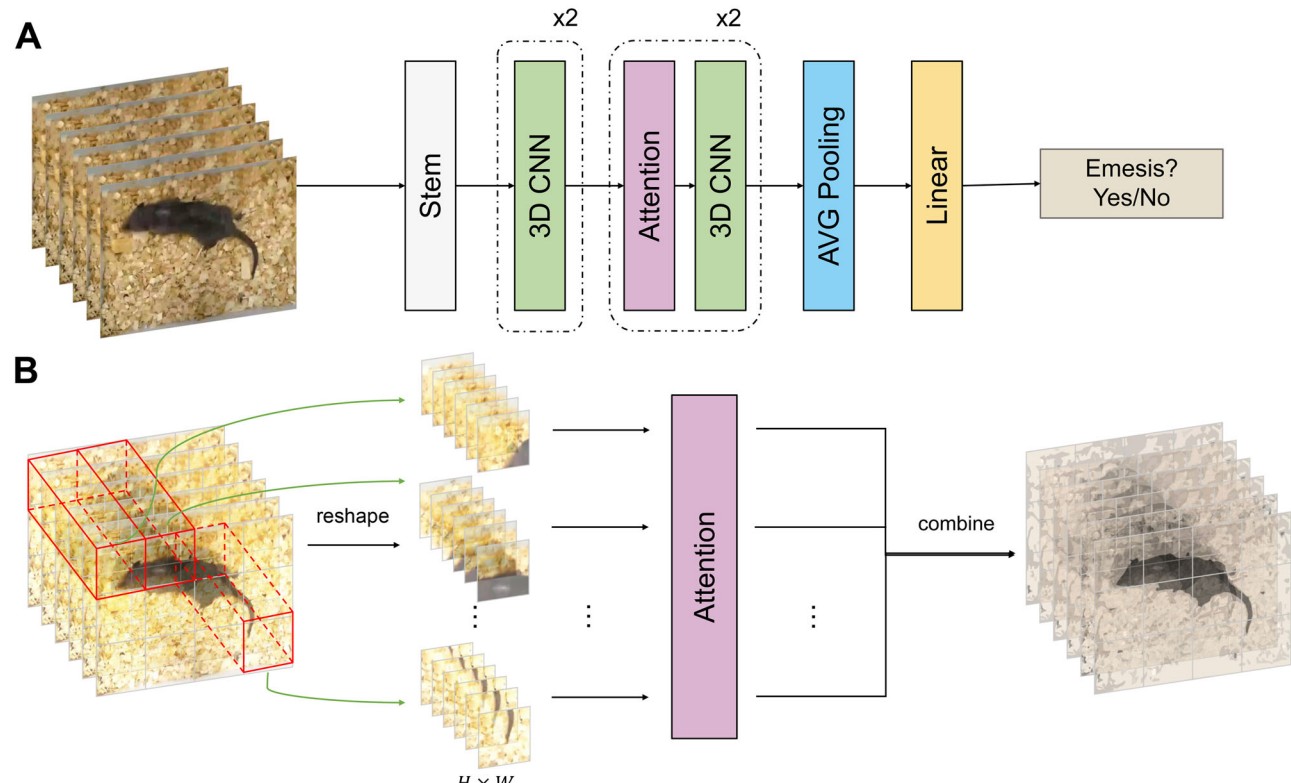

**Fig. 2 | Automatic Emesis Detection (AED) model architecture. A** Overview of the AED model. The core component consists of residual three-dimensional convolutional neural network (CNN) blocks and additional self-attention mechanisms. **B** Illustration of the self-attention mechanism.

respectively (Fig. 4B), underscoring the high reliability and accuracy of the model in identifying and quantifying emetic behaviours in *S. murinus* videos.

## Performance of the AED system in detecting emesis induced by various emetic drugs

To assess the ability of the AED system to recognise and quantify emesis induced by various emetic drugs in addition to motion-induced stimuli, we administered different emetic agents to *S. murinus*. We compared the quantification of emetic behaviour between the AED system and manual annotation.

Initially, we examined emesis recorded for short durations (≤30 min). In the RTX-induced emesis model, we observed $7.8 \pm 0.9$ episodes of R + V, and the AED system achieved a prediction accuracy of 100% (Fig. 5A, B). In the nicotine-induced emesis model, we observed $10.8 \pm 2.0$ episodes of R + V, and the AED system achieved a prediction accuracy of 100% (Fig. 5C, D). In the copper sulphate-induced emesis model, we observed $7.8 \pm 1.3$ episodes of R + V, and the AED system achieved a prediction accuracy of 100% (Fig. 5E, F). In the naloxone-induced emesis model, we observed $11.1 \pm 2.7$ episodes of R + V, and the AED system achieved a prediction accuracy of 97.10% (Fig. 5G, H). Finally, in the U46619-induced emesis model, we observed $12.3 \pm 1.1$ episodes of R + V, and the AED system achieved a prediction accuracy of 98.97% (Fig. 5I, J).

Subsequently, we evaluated emesis recorded for longer durations (2–6 h). In the cyclophosphamide-induced emesis model, we observed $16.1 \pm 4.9$ episodes of R + V, and the AED system achieved a prediction accuracy of 96.93% (Fig, 6A, B). In the exendin-4-induced emesis model, we observed $11.5 \pm 3.8$ episodes of R + V, and the AED system achieved a prediction accuracy of 98.91% (Fig. 6C, D).

## Application of the AED system in anti-emetic drug screening

In our final assessment, we examined the suitability of the AED system for screening anti-emetic drugs. To this end, we employed a cisplatin-

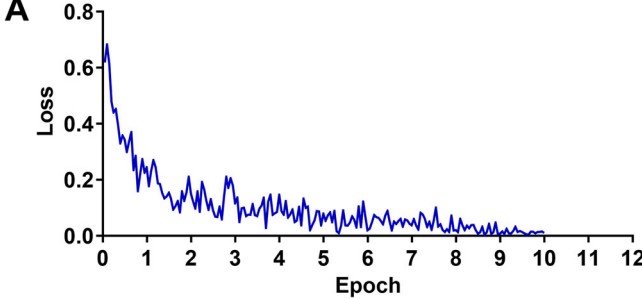

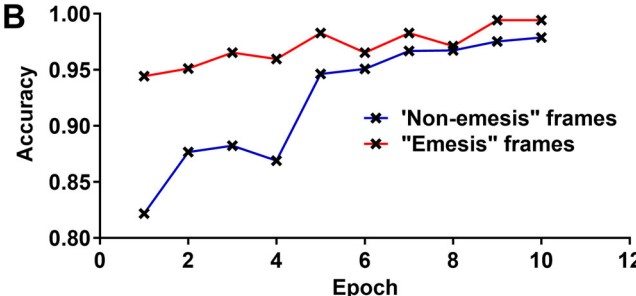

**Fig. 3 | Model design and training. A** Effects of different epochs on training loss. **B** Effects of different epochs on the accuracy of the model.

induced emesis model and performed pretreatment with palonosetron to validate the technique. Palonosetron (0.5 mg/kg, s.c.) or a saline vehicle (2 mL/kg, s.c.) was administered 0.5 h before the i.p. injection of cisplatin (30 mg/kg; Fig. 7A). Subsequently, emetic behaviour was recorded over a 6-h period.

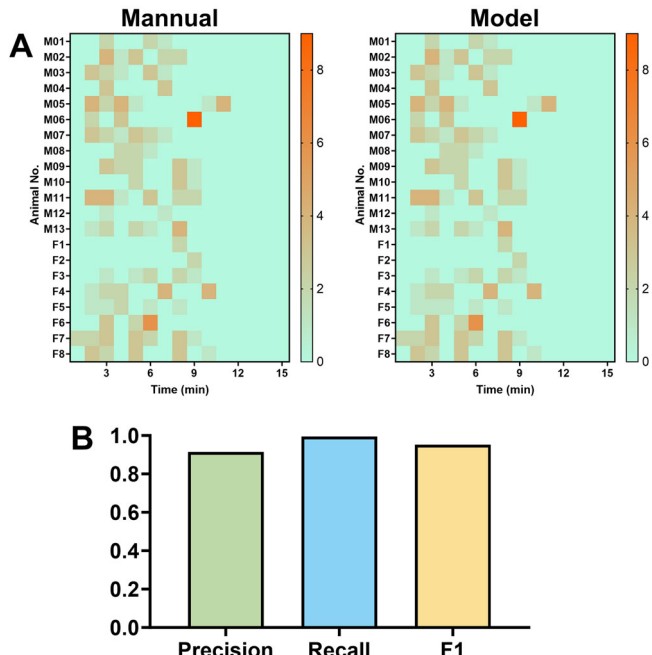

**Fig. 4 | Model evaluation using test datasets. A** Heatmap showing the latency and number of episodes of retching (R) + vomiting (V) per minute for each animal for motion-induced emesis (n = 21 animals). M = male, F = female. **B** The recall, precision, and F1 score of the trained model.

Manual quantification of emesis episodes revealed a 100% reduction in cisplatin-induced emesis following pretreatment with palonosetron (Fig. 7B, C). Quantification using the AED system produced similar results (Fig. 7D, E). These outcomes indicate that the AED system effectively detects alterations in emetic behaviour following anti-emetic drug administration, highlighting its potential utility in large-scale high-throughput screening studies of anti-emetic drugs.

## Performance of the AED system in the motion-induced emesis model of *Suncus murinus* in the observation box without bedding

To assess the robustness of the AED system, we further evaluated its performance using a motion-induced emesis model in *Suncus murinus* in an observation box devoid of bedding. In this experimental setup, four male and female *Suncus murinus* were subjected to motion stimulation. We recorded an average of 9.7 ± 4.4 episodes of R + V, with the AED system demonstrating a prediction accuracy of 100% (Supplementary Fig. 1).

## Comparison of the accuracy of emesis detection in *Suncus murinus* between AED and ActionFormer

As shown in Supplementary Table 2, the accuracy of emesis detection using ActionFormer in motion-, RTX-, nitcone-, copper sulphate-, naloxone-, U46619, cyclophosphamide-, exendin-4- and cisplatin-induced emesis model was 98.27%, 97.10%, 99.10%, 100%, 97.10%, 95.88%, 96.93%, 95.65% and 96.83%, respectively. AED outperforms ActionFormer on six out of nine metrics and achieves the same accuracy on other three metrics.

## Discussion

Manual quantification of emetic behaviours in *S. murinus* is arduous and time-consuming and may be prone to operator error. With rapid advancements in deep learning, automatic animal behaviour quantification tools with high accuracy have emerged. In this study, we pioneered the use of both 3D CNNs and self-attention mechanisms to develop the AED tool for the automatic detection of emesis, with an overall accuracy of 98.92% in *S. murinus*. Specifically, we used motion-induced emesis videos as training data, and the validation results from motion-induced emesis demonstrated an accuracy of 99.42%. In our model generalisation and application studies,

we evaluated the AED for its ability to detect emesis in response to various emetics, including RTX, nicotine, copper sulfate, naloxone, U46619, cyclophosphamide, exendin-4, and cisplatin. The prediction accuracy for these emetics was 97.10%, 100%, 100%, 97.10%, 98.97%, 96.93%, 98.91%, and 98.41%, respectively. Our findings indicate that the performance of the AED tool is on par with manual annotation across major emesis models and in anti-emetic drug screening studies of *S. murinus*. The exceptional prediction accuracy of the AED tool is likely attributable to high-quality and reproducible video recording, sufficient high-quality training data, appropriate design of the 3D CNN, and the optimisation of training parameters.

Various attempts have been made to devise automatic detection systems for emesis. Horn et al. conducted non-rigid tracking of *S. murinus* in videos to automatically detect emetic episodes; however, this method achieved an overall accuracy of only ~75%[31]. Furthermore, the combination of whole-body plethysmography with video recording has been employed as a semi-automated approach for emesis analysis in *S. murinus*[32]. Additionally, telemetric devices implanted into the abdominal wall of ferrets have enabled the recording of abdominal pressure, which was then used for automatic emesis detection[33]. Although these methods can achieve high accuracy, they often necessitate invasive surgeries and require expensive telemetric recording systems. In contrast, our study features a straightforward, reproducible, and cost-effective experimental setup, suggesting that the AED system holds promise for widespread adoption by researchers. Most importantly, our in vivo studies were conducted in accordance with the ARRIVE guidelines, which are aimed at enhancing the reproducibility of in vivo research. Therefore, the implementation of the AED tool in emesis research will standardise results across laboratories.

3D CNN evolved from 2D CNN to solve for video or 3D image analysis and is widely used in many fields[34,35]. In contrast to 2D CNN, 3D CNN is capable of learning a video's spatiotemporal characteristics, thus making up for the weakness of 2D CNN, which lacks temporal reasoning[34,35]. Additionally, by adding self-attention mechanisms and residual structure[34,35], our AED model was able to represent dependencies between inputs and outputs and was much easier to train. The emesis behaviour of *Suncus murinus* is characterised by its infrequency and short duration. To address these characteristics, we proposed a method that combines 3D convolution and attention mechanisms, leveraging their respective strengths in local and long-term sequence modelling, achieving extremely high accuracy.

For emesis frames not detected by the AED system, we observed various patterns across different treatments. In male *S. murinus* individuals treated with naloxone, we noted instances where emesis comprised only two retches, each with a lower frequency than typical retches (Supplementary Video 1). Similarly, emesis induced by naloxone in female *S. murinus* individuals consisted of four retches, also with a lower frequency than typical retches (Supplementary Video 2). In U46619-treated female *S. murinus* individuals, although emesis exhibited typical features with four retches, the frequency of retching was slightly lower than usual (Supplementary Video 3).

For animals treated with cyclophosphamide, three emetic events from one male were not detected by the AED system. The duration of these emetic events was very short, and the animal was in motion, walking and turning during the event (Supplementary Video 4–6). For exendin-4-treated animals, one emetic event was missed by the AED system. During this event, the amplitude of abdominal contractions during retching was significantly lower than usual (Supplementary Video 7). In the cisplatin study, a missed emetic event comprised seven retches, and the animal changed its location during the event (Supplementary Video 8).

These undetected emetic events can be categorised into three aspects: (1) fewer retches with longer intervals between each retch compared with typical emetic events; (2) a shorter duration of emesis with concurrent movement, such as walking or turning; and (3) a lower amplitude of abdominal contractions during each retch.

Additionally, the AED tool detected non-emesis frames, which included behaviours such as quick head shaking (Supplementary Video 9), which was often observed immediately after an episode of emesis (Supplementary

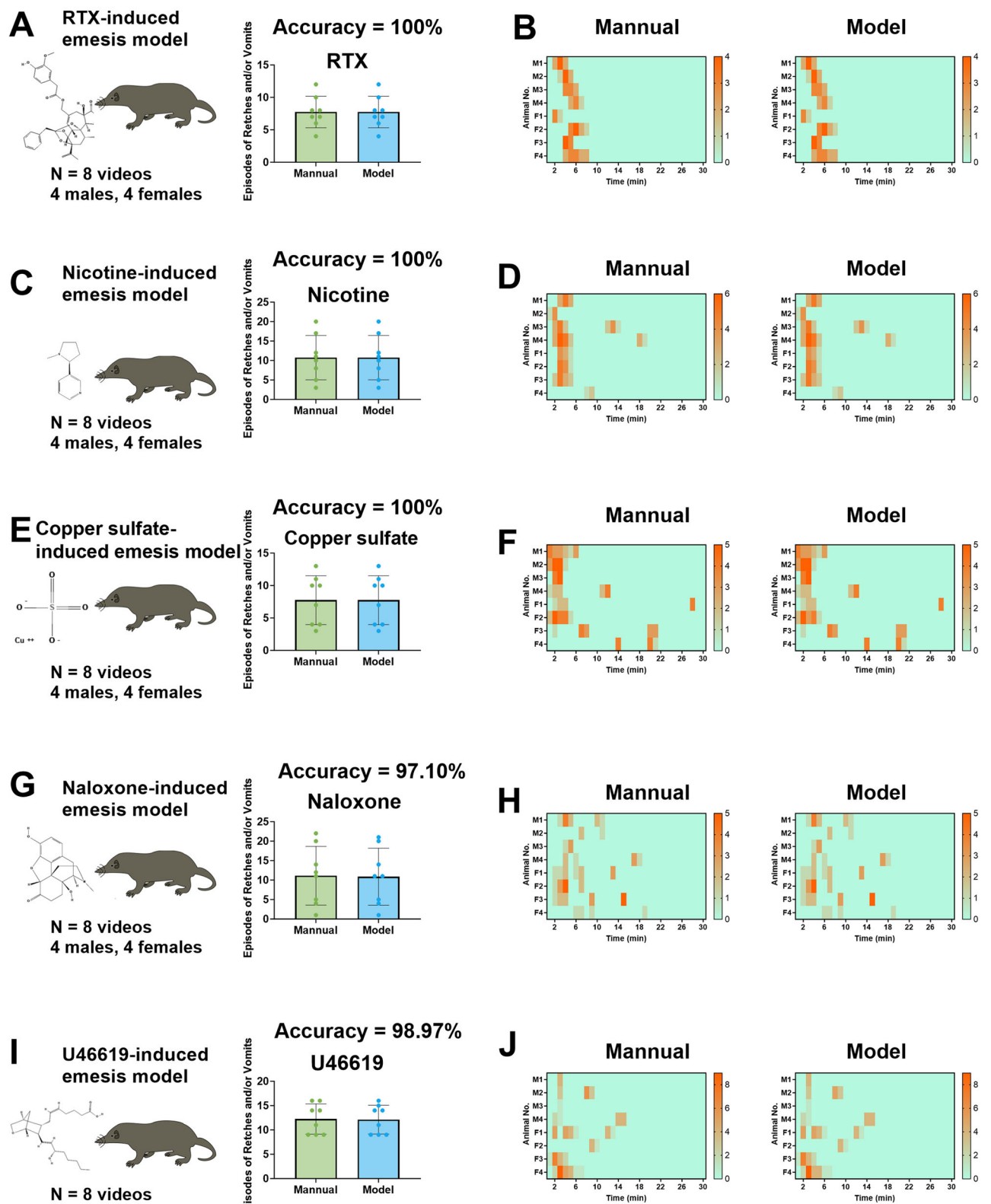

**Fig. 5 | Performance of the Automatic Emesis Detection (AED) system in short-duration (≤30 min) emesis models induced by six different drugs.** Episodes of retching (R) + vomiting (V) were quantified manually and using the AED system for the (**A**) resiniferatoxin (RTX)-induced, (**C**) nicotine-induced, (**E**) copper sulfate-induced, (**G**) naloxone-induced, and (**I**) U46619-induced emesis models. Heatmaps showing the latency and number of episodes of R + V per minute for each animal in the (**B**) RTX-induced, (**D**) nicotine-induced, (**F**) copper sulfate-induced, (**H**) naloxone-induced, and (**J**) U46619-induced emesis models. Data are presented as the mean ± standard deviation (n = 8 animals). M = male, F = female. The 2D structures of corresponding compounds were obtained from the PubChem database (https://pubchem.ncbi.nlm.nih.gov/).

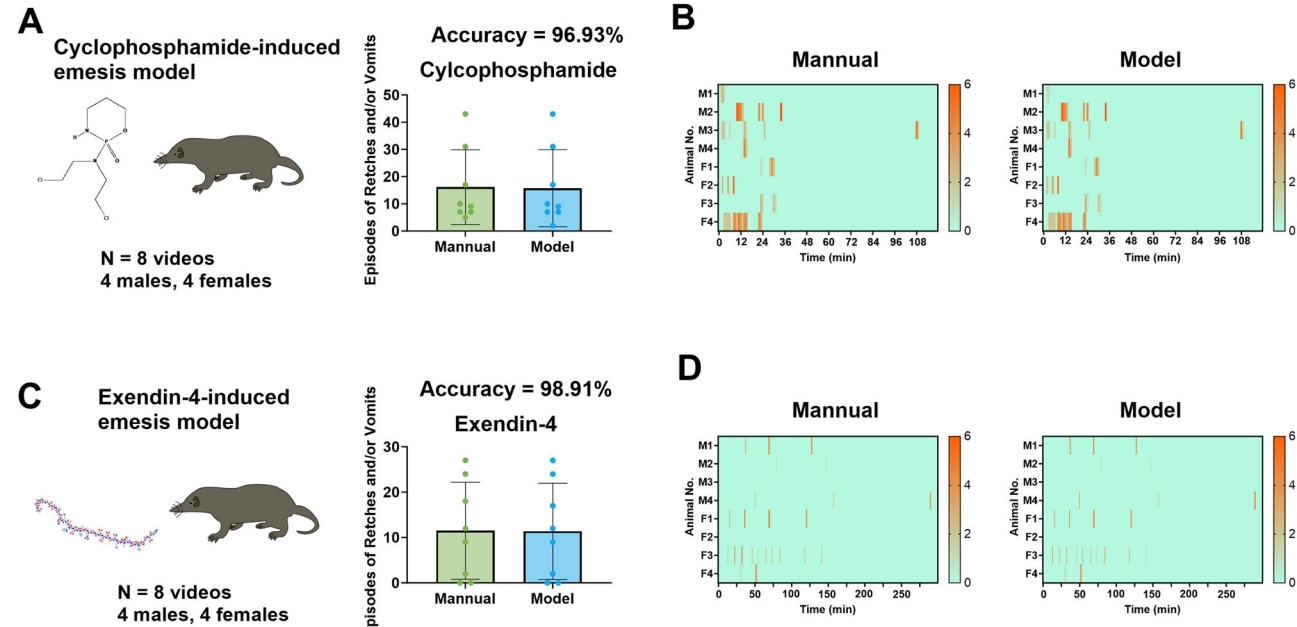

**Fig. 6 | Performance of the Automatic Emesis Detection (AED) system in long-duration (≥2 h) emesis models induced by six different drugs.** Episodes of retching (R) + vomiting (V) quantified manually and using the AED system in the (**A**) cyclophosphamide-induced and (**C**) exendin-4-induced emesis models. Heat-maps showing the latency and number of episodes of R + V per minute for each animal in the (**B**) cyclophosphamide-induced and (**D**) exendin-4-induced emesis models. Data are presented as mean ± standard deviation (n = 8 animals). M = male, F = female. The 2D structures of corresponding compounds were obtained from the PubChem database (https://pubchem.ncbi.nlm.nih.gov/).

Video 10). The inclusion of such frames in the training dataset may lead to incorrect detection using our model. Another non-emetic behaviour detected was back-and-forth body movement, which resembled emetic behaviour (Supplementary Video 11). Chewing with rapid nose movement (Supplementary Video 12) and using the forelimbs to rub the face with rhythmic head movements (Supplementary Video 13) were also detected, as these resembled the rhythmic movement during emesis. In naloxone-treated animals, trembling behaviours of the body were also falsely detected as emetic behaviours (Supplementary Video 14). These falsely detected non-emetic behaviours should be used as negative frames for further training to reduce the rate of false negative results.

In our study, we employed a 3D CNN to accurately detect emesis behaviours in animals during long video recordings. One challenge in this context is differentiating between emesis and other behaviours, such as grooming and rearing, which can occur frequently and may be visually similar in the footage. Grooming behaviours typically involve rhythmic movements and can include licking and scratching[36], while rearing involves the animal standing on its hind legs[37]. Our 3D CNN was specifically trained to recognise the unique spatiotemporal patterns associated with emesis, enabling it to distinguish these behaviours effectively. The model leverages the temporal aspect of the video data, analysing sequences of frames to identify the rapid, characteristic motions of emetic event. In our study, we also compared the accuracy of emesis detection in *Suncus murinus* between AED and ActionFormer. ActionFormer is a localisation model designed to detect and classify actions in videos by leveraging temporal attention and representation learning[38]. It excels in tasks requiring fine-grained temporal segmentation, making it well-suited for scenarios with continuous or overlapping activities. The comparison between AED and ActionFormer highlights the strengths of the developed model in detecting emesis in *Suncus murinus*. By framing the problem as a binary classification task instead of action localisation, AED achieves remarkable accuracy across various stimuli, consistently outperforming or matching the performance of ActionFormer. Notably, AED's design effectively addresses the unique challenges of emesis detection, such as its infrequent occurrence and short duration, without requiring extensive modifications or additional computational resources.

The accuracy of the AED tool can be improved. To enhance the accuracy in detecting emesis, incorporating additional training datasets featuring various emetic stimuli may be beneficial. Furthermore, to minimise false positives identified in the videos, future studies should consider labelling the 'start' and 'end' of emetic behaviour on a frame-by-frame basis. The use of this technology to characterise behaviours around emetic episodes that may indicate 'nausea' is now being explored.

Compared with manual analysis, employing deep learning-based automatic analysis improves efficiency and accuracy and may also mitigate human bias and errors. In our investigation, we devised the AED system for the automated quantification of emesis in *S. murinus*. Furthermore, our study provides valuable insights into the development of deep learning neural network models aimed at automating the analysis of various other behaviours in *S. murinus*.

## Methods
### Animals
Male (~60–80 g, 13–17 weeks old) and female (~35–45 g, 13–17 weeks old) *S. murinus* (House Musk Shrew) were procured from The Chinese University of Hong Kong. They were housed in a temperature-controlled room set at 24 ± 1 °C with artificial lighting from 0600 to 1800 h. Humidity was maintained at 50 ± 5%. Water and dry cat chow pellets (Feline Diet 5003; PMI Feeds, St. Louis, MO, USA) were freely accessible. All experimental procedures were conducted under the authorisation of the Government of the Hong Kong SAR and were approved by the Animal Experimentation Ethics Committee of The Chinese University of Hong Kong (Approved No.: 23-246-MIS). Animal studies are reported in compliance with the ARRIVE guidelines[39].

### Motion-induced emesis model
Motion-induced emesis in *S. murinus* was induced according to established protocols[40]. In brief, the animals underwent a 1-h habituation period in the observation chamber prior to exposure to provocative motion. The motion, generated by a shaker (Heidolph Promax; Heidolph Instruments, Schwabach, Germany), comprised a horizontal displacement of 4 cm at a frequency of 1 Hz, for 15–20 min.

**Fig. 7 | Application of the Automatic Emesis Detection (AED) system in a drug screening paradigm. A** Diagram showing the experimental design of an anti-emetic test. **B**, **C** Quantification of emetic behaviour in palonosetron (Palo)-treated and saline (Sal)-treated groups using manual annotation. **B** Episodes of retching (R) + vomiting (V), and (**C**) heatmap showing the latency and number of episodes of R + V per minute for each animal are shown. **D**, **E** Quantification of emetic behaviour in the palonosetron (Palo)-treated and saline (Sal)-treated groups using the AED system. **D** Episodes of R + V and (**E**) heatmap showing the latency and number of episodes of R + V per minute for each animal are shown. Data are presented as the mean ± standard deviation (n = 5 animals). **P < 0.05 (unpaired Student's t-test).

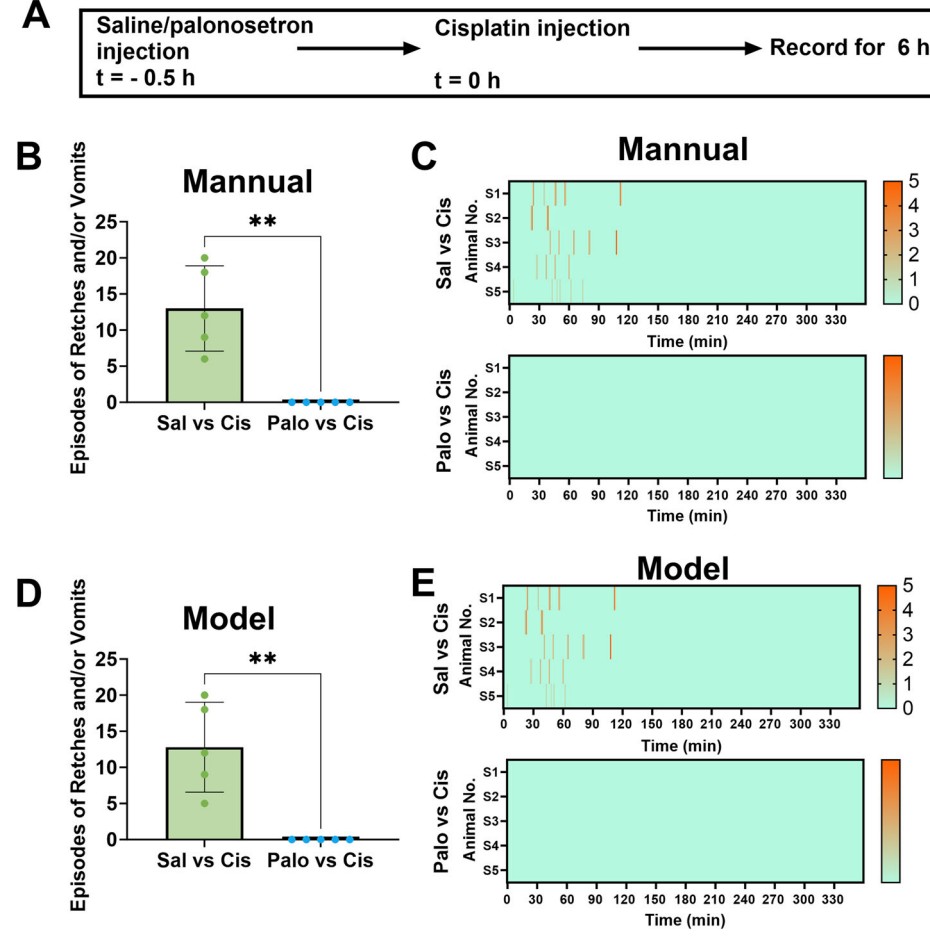

## Behavioural recording

Emetic behaviour was documented using a Logitech (Lausanne, Switzerland) C930e webcam connected to a Microsoft desktop running Windows 10 Pro (Microsoft Corporation, Redmond, WA, USA). Video recording was achieved using Logitech Capture 2.08.11 software. The recording settings were as follows: contrast, 0; brightness, 170; resolution, 1920 × 1080; and frame rate, 30 fps.

## Annotation of emetic behaviour in *S. murinus*

The start of an emetic episode was identified as the moment when an *S. murinus* individual commenced retching and/or vomiting. The end of such an episode was identified as when the animal finished its final retching or vomiting event. For manual annotation, every video underwent thorough examination at a standard speed (1×, 30 fps). The commencement and cessation times of each emetic episode were manually marked, cross-checked and verified by three experienced observers. Frames occurring during an emetic episode were classified as 'emesis' frames, while those outside such episodes were labelled as 'non-emesis' frames.

## Data collection and preprocessing

For manually annotated emesis videos, our initial step involved resizing to a standardised dimension of 360*640. Following this, we extracted 30 frames from every 1-s video clip labelled with emetic episodes as 'emesis' frames and inputted 30 sequential frames into the neural network for training, while maintaining a frame rate of 30 fps to facilitate subsequent training. The input images were then split into two subsets: a training set and a testing set, maintaining an ~9:1 ratio by randomly assigning the motion-stimulated animals to either the training or testing set. The training and testing are done on different animals. All images labelled as emesis frames were categorised as positive samples. We extracted images labelled as non-emesis frames

from 100 videos at uniform time intervals, with each negative video sample comprising 30 frames, to function as negative samples.

## Attention-Augmented R3D model architecture

Our 3D CNN was adopted from the model described by Tran et al.[29], facilitating fully temporal and spatial information. Additional self-attention mechanisms were used to extract global dependencies of the inputs. The network architecture consisted of four CNN blocks and two additional self-attention modules (Fig. 2A). Four-dimensional convolution filters were used. After the features were extracted by the previous layers, the average pooling layer was used to reduce the dimensionality and simplify the computation. Finally, a fully connected layer returned a binary classification result to determine whether the animal was vomiting in the input frame. Figure 2B demonstrates the information interactions inside the self-attention architecture. The input frames were transformed into feature maps using 3D convolutions, and then reshaped along the temporal axis. The vectors were projected by multiple heads of the self-attention module into several information spaces, enabling the model to capture changes in animal motion in input frames at different timepoints.

## Model training and testing

This model was implemented using PyTorch. In the training phase, we performed 20,000 iterations with a batch size of 24. Each data sample consisted of 30 sequential video frames, which underwent augmentation through data balancing techniques. As described above, the number of emesis frames was much less than the number of non-emesis-frames, that is, the number of negative samples was much greater than the number of positive samples. To balance the number of positive and negative samples, we sampled the training data using a 1:1 ratio. The training process used

hardware equipped with a 32GB RAM NVIDIA GeForce V100 GPU (NVIDIA, Santa Clara, CA, USA).

### Model evaluation/generalisation using various emetic drugs

For model evaluation and generalisation for emesis induced by various emetic stimuli, the data preprocessing mirrored that of the training dataset. Each frame was fed into the model individually, allowing us to obtain predictions of emesis or non-emesis for each frame. The clips used for prediction sampled at 1-frame interval were shifting through the full video for detecting emetic events. To evaluate performance, we calculated recall (the number of correctly predicted 'emesis' frames divided by the number of manually annotated 'emesis' frames), precision (the number of correctly predicted 'emesis' frames divided by the total number of predicted 'emesis' frames), and F1 score (2*recall*precision/(recall + precision)).

For emetic drug testing, four male and four female *S. murinus* individuals were used for each drug. On the day of the experiments, the animals were acclimated in the observation chamber for 1 h. Subsequently, the animals were administered RTX (100 µg/kg, subcutaneous [s.c.])[20], nicotine (5 mg/kg, s.c.)[21], copper sulphate (120 mg/kg, oral)[22], naloxone (60 mg/kg, s.c.)[23], U46619 (300 µg/kg, s.c.)[24], cyclophosphamide (200 mg/kg, intraperitoneal [i.p.])[25], or exendin-4 (30 nmol/kg, s.c.)[26,27]. Emetic behaviour following the administration of the aforementioned emetic drugs was recorded for 30, 30, 20, 30, 30, 120, and 360 min, respectively. Recall was used to assess the accuracy of the AED tool in emetic drug testing experiments.

### Model validation using an anti-emetic drug experiment

Four male and six female *S. murinus* individuals were randomly allocated into two treatment groups. On the day of the experiment, the animals were permitted to acclimate in the observation chamber for 1 h. Subsequently, they were injected with palonosetron (0.5 mg/kg, s.c.; three males and two females) or vehicle (saline, 2 mL/kg, s.c.; two males and three females) 30 min prior to cisplatin (30 mg/kg, i.p.) injection[22,28]. Emetic behaviour was then assessed for 6 h. Recall was used to assess the accuracy of AED tool in the anti-emetic drug experiment.

### Emesis detection in *Suncus murinus* using ActionFormer

To ensure methodological fairness and accommodate our computational budget, the number of parameters was reduced in ActionFormer to align its training costs with those of our model. Regarding the training data, due to the relative scarcity of positive samples, the videos were segmented into clips with varying start times to generate multiple training instances containing positive data points. We adhered to the action localisation protocol for training ActionFormer[38], subsequently modifying the number of action classes to a single category emesis. During testing, a prediction is deemed positive if ActionFormer's predicted temporal window overlaps with the ground truth, which advantages ActionFormer. For both training and testing phases, we input a 20-s sequence into the model at once. Given the differences in model architectures, we adjusted the number of training steps for ActionFormer and selected the model that demonstrated the best performance on the validation set for comparative analysis.

### Drug formulation

11α,9α-Epoxymethano-15*S*-hydroxyprosta-5*Z*,13*E*-dienoic acid (U46619; Cayman Chemical Company, Ann Arbor, MI, USA) was prepared in absolute ethanol at 1 mg/mL and stored at −20 °C. It was diluted using saline immediately prior to use (final ethanol concentration, 15% v/v). Copper sulphate pentahydrate (Riedel-DeHaën, Seelze, Germany) was dissolved in distilled water. RTX (Sigma, St Louis, MO, USA) was dissolved in Tween 80/ethanol/saline (0.9% w/v) at a ratio of 1:1:8. (−)-Nicotine di-D-tartrate (Research Biochemicals International, Natick, MA, USA), exendin-4 (Tocris Bioscience, Bristol, UK), naloxone hydrochloride (Sigma), palonosetron hydrochloride (Toronto Research Chemicals, Toronto, Canada), and cyclophosphamide monohydrate (Sigma) were dissolved in saline. Cisplatin (Merck, Darmstadt, Germany) was dissolved in acidified saline (154 mM NaCl, adjusted to pH 4.0 with 0.1 M HCl). All drugs except cisplatin were administered at 2 mL/kg unless stated otherwise. Cisplatin was administered at 10 mL/kg. Doses are expressed as the free base weight unless otherwise indicated.

### Statistical analysis and reproducibility

Data were analysed using Prism 9.0 (GraphPad Software, San Diego, CA, USA). The results are presented as mean ± standard deviation. Details of sample sizes are provided in the figure legends or corresponding figure panels. Differences in means between two groups were assessed using an unpaired two-tailed Student's t-test. Results were deemed significant when $P < 0.05$.

### Reporting summary

Further information on research design is available in the Nature Portfolio Reporting Summary linked to this article.

## Data availability

The training and test videos generated during the current study can be downloaded from Harvard Dataverse for short videos (https://doi.org/10.7910/DVN/XXAWEL)[41] and ScienceDB for long videos (https://doi.org/10.57760/sciencedb.15205)[42].

## Code availability

The codes for model training and test can be downloaded from Zenodo (https://doi.org/10.5281/zenodo.14591163)[43].

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

## Acknowledgements
Professional English language editing support provided by AsiaEdit (asiaedit.com). This study was supported by funding from by the Emesis Research Group and School of Biomedical Sciences, The Chinese University of Hong Kong.

## Author contributions
J.A.R. and Z.L. conceptualised the study; Z.L. performed the experiments; Z.L., Y.Q., X.H., D.C. and Z.H. analysed the data; J.L., M.N., L.L., Z.H., Z.T.L., L.Y., A.K., Y.D., S.W.C. and L.T. reviewed and finalised the manuscript; J.A.R., Z.L. and Y.Q. wrote the manuscript. All of the authors have approved the manuscript for submission.

## Competing interests
The authors declare no competing interests.

## Ethical approval
All animal care and experimental procedures were conducted under license from the Government of the Hong Kong SAR and the Animal Experimentation Ethics Committee of The Chinese University of Hong Kong (Approved No.: 23-246-MIS). We have complied with all relevant ethical regulations for animal use.
