## [Peer Review file · Communications Biology]

A deep learning-based system for automatic detection of emesis with high accuracy in *Suncus murinus*

Corresponding Author: Professor John Rudd

Version 0:

Reviewer comments:

Reviewer #1

(Remarks to the Author)

Overall - this is a useful paper by Dr. Rudd and colleagues and I support its publication if the following things can be clarified.

The DL side of things has a couple oddities. It appears sound, but could use some clarifications:

Authors use a 3D network, but state multiple times that their examples have labels attached to frames. How many sequential frames are provided to the network (e.g. temporal receptive field)?

Clarify the statement "we converted the annotated videos into images ...". see comment on receptive field above.

How are full videos predicted on for detecting events? Specifically, how are clips selected to be passed through the network. Are clips used for prediction sampled at 1-frame intervals shifting through the video, or are they somehow aligned to ground truth annotation?

Authors describe event performance, but don't have any description in methods how events are compared.

Confirm that performance metrics are on a frame wise basis.

Provide a confusion matrix.

Training-validation split needs more details. Is the 9:1 split random or stratified by some grouping (e.g. event or animal)? One of the figures suggest stratified by animal, but it's never explicitly stated. (It's worth noting that the text suggests that an adequate test set was used: "only motion induced was used to train".)

Confirm that training and testing are done on different animals.

Figure 4A - please add two heat maps as in figure 5B. One for ground truth the other for prediction from NN.

Geuther et. al. use 3D CNN for grooming behavior classification in 2021. DOI: <https://doi.org/10.7554/eLife.63207> we needed a large number of training videos for the network to generalize to genetically diverse mice. This paper should be cited.

Figure 6 legend says it short Short-duration, but these are 2hr videos (long or medium).

The motion induced emesis assay - is the motion induced in the recording chamber? or is the animal removed, motion is applied, and then its placed back in the chamber?

For long videos, the animals must have grooming and rearing, but the network does not confuse these with emesis. It would be interesting to discuss.

Reviewer #2

(Remarks to the Author)

Overview

The authors developed a tool to automatically detect emesis in *Suncus murinus* from video recordings of their behaviors in a common laboratory cage. Their tool involves an artificial neural network combining convolutional layers and attention modules. The model was trained using annotations created by human observation. The accuracy reached more than 98% in detecting emesis elicited by administrations of different types of compounds.

Although the tool is undoubtedly valuable in analyzing the emesis of *Suncus*, its novelty and impact are strictly limited to this purpose. Their neural network appears to use a standard architecture, and the conceptual advancement is unclear in its design and application. The model involves common convolutional layers and attention mechanisms; their task is standard temporal action detection. Their evaluation is specifically limited to the emesis of *Suncus*, not to any other behavior or other species. Their discussion and future perspectives were also dedicated to further enhancement in the detection of emesis of *Suncus murinus*, representing their scope, which is indeed sharply focused on the emesis of *Suncus*.

Major points

(line 95) The author claims that the detection of emesis imposes unique challenges in computer vision and that they have offered conceptually novel architecture to solve these issues. However, this claim is not convincing based on the supplementary videos and the architecture of the neural network. Rather, it appeared to be a relatively easy task as the actions to be detected were relatively homogenous, the animal body occupied many pixels, and the background was consistent across videos. My concern is that the good performance was due to the fine-tuning of model parameters (such as factors related to dimension reduction of the temporal axis in the model) rather than the conceptual design of model architecture. If so, the advancement made in the present study is specifically about detecting *Suncus* emesis (i.e., local optimization), which perhaps lacks broader interest.

To ensure impact and interest for the broader community, authors may want to 1) test their tool with common benchmark datasets and 2) compare it with other models using the *Suncus* dataset.

The performance of their tool should be tested using common benchmark datasets (e.g., THUMOS'14 and ActivityNet-1.3), and then compared with existing models (e.g., ActionFormer, Re2TAL, and AdaTAD, which can be trained from scratch).

Benchmark scores for related studies: <https://paperswithcode.com/task/action-recognition>

ActionFormer: <https://doi.org/10.48550/arXiv.2202.07925>

Re2TAL: <https://doi.org/10.48550/arXiv.2211.14053>

AdaTAD: <https://doi.org/10.48550/arXiv.2311.17241>

Then, the performance of the above tools can also be tested to detect *Suncus* emesis. Several parameters were also tunable in these models. If the developed tool shows superior performance, it supports their main claim that emesis detection involves unique challenges (infrequent and short actions), and their model provides advanced solutions.

Minor points

(Line 166 and Fig.2) I felt that the word "batchfy" is confusing as "batch" is used in a different sense in artificial neural networks (as the authors did in line 172). In the vision transformer, it is termed patch partitioning and embedding.

(Line 246) The authors should solidify the quality of their dataset. It was unclear how skilled the human annotators were in the present format. The high score by the neural network can be achieved by systematically biased annotations. So, the accuracy of human annotation should be tested through inter-observer reliability.

(Line 444) Depending on the response to the previous comments, it seems that the main contribution of this paper is not conceptual advancement, but rather the implementation of the automated detection system for *Suncus* emesis and the development of its training dataset. If this is the case, the training dataset and the developed system should be made publicly available through cloud-based repositories. Ensuring easy access to these resources, not only through the manuscript but also by making the data and tools available, can significantly contribute to the research community. Ideally, the code should include a demo script for the training and evaluation to help users get started.

Reviewer #3

(Remarks to the Author)

In this manuscript, Rudd et al developed a deep learning-based system for detecting emesis behaviors in *Suncus murinus* with high accuracy. Previous research analyzing emesis behaviors in *S. murinus* relies on manual scoring, which is laborious. Rudd et al demonstrates that their model can be used to efficiently and accurately score emesis behaviors in several conditions, including monition and with 8 different emetic drugs in short and long video durations. Together, this work addresses an unmet need in this field, and has great potential to be used by other researchers investigating emesis and drug screens.

I find this study well designed and conducted. The authors discussed most caveats and improvements that they could address in future works. For example, the accuracy decreases in longer videos, and the short emetic episodes can be missed. I have a few questions that just need to be clarified:

1. How reliable are the AED scoring? If you run the same testing video a few times, do you get the same results, especially

for those long videos, or videos containing very short emetic episodes/body trembling?

2. For the future work, if you train AED with all the videos including motion and 8 drug conditions, will it further improve the model to address the false negatives?

3. Will it work well if the video parameters are changed, for example, no bedding? How robust is the model to be used in a different lab video setting?

4. Deep learning-based systems for behavioral analysis has become more and more useful in the field of neuroscience. The authors also cited many key works (ref 11-15). In all cases, the training data and/or codes are publicly available for open access by other researchers for scrutiny, improvement and use. Is this what the authors are also considering? It will certainly be a great tool for the field.

Chuchu Zhang

Version 1:

Reviewer comments:

Reviewer #2

(Remarks to the Author)

I thank the authors for revising the manuscript. They have adequately addressed all of the concerns I raised in the previous review. I have only one remaining comment.

The authors have compared the accuracy of AED (proposed) with ActionFormer (an existing tool) for action recognition. The results demonstrated that their method outperformed ActionFormer on six out of nine tests and was comparable to the rest. However, the details of these results and the corresponding methods and discussion are missing from both the main text and supplemental materials. So, I recommend that these details be included in the paper to provide a more convincing evaluation of the proposed method.

Reviewer #3

(Remarks to the Author)

This manuscript is a great contribution to the research community. The authors have addressed all my questions, and I have no more questions after the revision.

Reviewer #4

(Remarks to the Author)

Dr. Rudd and colleagues have detailed an automated method for detecting emesis in *Suncus murinus* using a 3D CNN, reporting its high performance based on widely accepted measures of the field (precision, accuracy, F1beta). The generalization across several models of emesis is commendable, as is the pharmacological validation with palonosetron/cisplatin co-administration. Given *Suncus murinus*' place as a premier model organism for studying emesis, and the publicly available data set and model, this tool promises to further the field by relieving the burden of manual emesis scoring.

As a replacement to reviewer 1, I find the revision of this paper to adequately address the points previously highlighted, and think the manuscript is in a condition worthy of publication.

Reviewers' comments:

Reviewer #1 (Remarks to the Author):

Overall - this is a useful paper by Dr. Rudd and colleagues and I support its publication if the following things can be clarified.

The DL side of things has a couple oddities. It appears sound, but could use some clarifications:

Authors use a 3D network, but state multiple times that their examples have labels attached to frames. How many sequential frames are provided to the network (e.g. temporal receptive field)?

Response: Thank you for the comments. We provided 30 sequential frames from 1 sec video clip annotated with emesis to the network for training.

Revised text: page 7, line 150-152: "Following this, we extracted 30 frames from every 1-second video clip labelled with emetic episodes as 'emesis' frames and input 30 sequential frames into the neural network for training."

Clarify the statement "we converted the annotated videos into images ...". see comment on receptive field above.

Response: We have clarified this in the revised manuscript. We extracted 30 frames from every 1-second video clip labelled with emetic episodes as 'emesis' frames and inputted 30 consecutive frames into the neural network for training.

Revised text: page 7, line 150-152: "Following this, we extracted 30 frames from every 1-second video clip labelled with emetic episodes as 'emesis' frames and inputted 30 sequential frames into the neural network for training."

How are full videos predicted on for detecting events? Specifically, how are clips selected to be passed through the network. Are clips used for prediction sampled at 1-frame intervals shifting through the video, or are they somehow aligned to ground truth annotation?

Response: Thank you for the comments. We have clarified this in the revised manuscript. The clips used for prediction sampled at 1-frame interval are shifting through the video for detecting emetic events.

Revised text: page 8, line 189-191: “The clips used for prediction sampled at 1-frame interval were shifting through the full video for detecting emetic events.”

Authors describe event performance, but don't have any description in methods how events are compared.

Response: Thank you for the comments. We have provided details for describing event performance in the revised methods.

Revised text: page 8, line 191-194: “To evaluate performance, we calculated recall (the number of correctly predicted ‘emesis’ frames divided by the number of manually annotated ‘emesis’ frames), precision (the number of correctly predicted ‘emesis’ frames divided by the total number of predicted ‘emesis’ frames), and F1 score ($2 * \text{recall} * \text{precision} / (\text{recall} + \text{precision})$).”

Confirm that performance metrics are on a frame wise basis.

Response: Yes, the performance metrics are on a frame wise basis. We evaluate the performance of this tool by determining whether it can detect emesis in each video clip (1 second). If emetic event is present, the frames from that second are classified as ‘emesis’ frames. If no ‘emesis’ is detected in that second clip, the frames are classified as ‘non-emesis’ frames.

Provide a confusion matrix.

Response: We have provided a confusion matrix in the revised manuscript (See supplemental Table S1).

Revised text: page 13, line 319: “The confusion matrix of the test dataset was shown in Supplemental Table S1.”

Training-validation split needs more details. Is the 9:1 split random or stratified by some grouping (e.g. event or animal)? One of the figures suggest stratified by animal, but it's never explicitly stated. (It's worth noting that the text suggests that an adequate test set was used: "only motion induced was used to train".)

Response: The selection of training and validation split is random. We have clarified this in the revised manuscript. The input images were then split into two subsets: a training set and a test set, maintaining an approximate 9:1 ratio by randomly assigning the motion-stimulated animals to either the training or testing set.

Revised text: page 7, line 153-156: “The input images were then split into two subsets: a training set and a testing set, maintaining an approximate 9:1 ratio by randomly assigning the motion-stimulated animals to either the training or testing set.”

Confirm that training and testing are done on different animals.

Response: We confirm that training and testing are done on different animals.

Revised text: page 7, line 157: “The training and testing are done on different animals.”

Figure 4A - please add two heat maps as in figure 5B. One for ground truth the other for prediction from NN.

Response: We have added two heatmaps in Figure 5B as suggested by the reviewer.

Geuther et. al. use 3D CNN for grooming behavior classification in 2021. DOI: <https://doi.org/10.7554/eLife.63207> we needed a large number of training videos for the network to generalize to genetically diverse mice. This paper should be cited.

Response: We have cited this reference as suggested by the reviewer.

Revised text: page 4, line 94-96: “Geuther et al. used 3D CNN for grooming behavior classification, and this tool can detect grooming behavior with human observer-level performance [15].”

Figure 6 legend says it short Short-duration, but these are 2hr videos (long or medium).

Response: We are sorry for the mistake. It should be long duration, and we have corrected this in the revised manuscript.

Revised text: page 23, line 619-620: “Figure 6. Performance of the Automatic Emesis Detection (AED) system in long-duration (≥ 2 hours) emesis models induced by six different drugs.”

The motion induced emesis assay - is the motion induced in the recording chamber? or is the animal removed, motion is applied, and then its placed back in the chamber?

Response: Thank you for the comments. For the motion-induced emesis assay, the recording chamber is placed on a motion platform (a shaker), and it moves with the shaker, achieving a horizontal displacement of 4 cm at a frequency of 1 Hz. The animals were in the recording chamber during a 1-hour habituation period and while motion was applied.

For long videos, the animals must have grooming and rearing, but the network does not confuse these with emesis. It would be interesting to discuss.

Response: We have further discussed this issue as suggested by the reviewer.

Revised text: page 18-19, line 452-460: “In our study, we employed a 3D CNN to accurately detect emesis behaviors in animals during long video recordings. One challenge in this context is differentiating between emesis and other behaviors, such as grooming and rearing, which can occur frequently and may be visually similar in the footage. Grooming behaviors typically involve rhythmic movements and can include licking and scratching [33], while rearing involves the animal standing on its hind legs [34]. Our 3D CNN was specifically trained to recognize the unique spatiotemporal patterns associated with emesis, enabling it to distinguish these behaviors effectively. The model leverages the temporal aspect of the video data, analyzing sequences of frames to identify the rapid, characteristic motions of emetic event.”

Reviewer #2 (Remarks to the Author):

Overview

The authors developed a tool to automatically detect emesis in *Suncus murinus* from video recordings of their behaviors in a common laboratory cage. Their tool involves an artificial neural network combining convolutional layers and attention modules. The model was trained using annotations created by human observation. The accuracy reached more than 98% in detecting emesis elicited by administrations of different types of compounds.

Although the tool is undoubtedly valuable in analyzing the emesis of *Suncus*, its novelty and impact are strictly limited to this purpose. Their neural network appears to use a standard architecture, and the conceptual advancement is unclear in its design and application. The model involves common convolutional layers and attention mechanisms; their task is standard temporal action detection. Their evaluation is specifically limited to the emesis of *Suncus*, not to any other behavior or other species. Their discussion and future perspectives were also dedicated to further enhancement in the detection of emesis of *Suncus murinus*, representing their scope, which is indeed sharply focused on the emesis of *Suncus*.

Major points

(line 95) The author claims that the detection of emesis imposes unique challenges in computer vision and that they have offered conceptually novel architecture to solve these issues. However, this claim is not convincing based on the supplementary videos and the architecture of the neural network. Rather, it appeared to be a relatively easy task as the

actions to be detected were relatively homogenous, the animal body occupied many pixels, and the background was consistent across videos. My concern is that the good performance was due to the fine-tuning of model parameters (such as factors related to dimension reduction of the temporal axis in the model) rather than the conceptual design of model architecture. If so, the advancement made in the present study is specifically about detecting *Suncus emesis* (i.e., local optimization), which perhaps lacks broader interest.

To ensure impact and interest for the broader community, authors may want to 1) test their tool with common benchmark datasets and 2) compare it with other models using the *Suncus* dataset.

The performance of their tool should be tested using common benchmark datasets (e.g., THUMOS'14 and ActivityNet-1.3), and then compared with existing models (e.g., ActionFormer, Re2TAL, and AdaTAD, which can be trained from scratch).

Benchmark scores for related studies: <https://paperswithcode.com/task/action-recognition>

ActionFormer: <https://doi.org/10.48550/arXiv.2202.07925>

Re2TAL: <https://doi.org/10.48550/arXiv.2211.14053>

AdaTAD: <https://doi.org/10.48550/arXiv.2311.17241>

Then, the performance of the above tools can also be tested to detect *Suncus emesis*. Several parameters were also tunable in these models. If the developed tool shows superior performance, it supports their main claim that emesis detection involves unique challenges (infrequent and short actions), and their model provides advanced solutions.

Response: Thank you for the comments. We want to clarify that we formulate this emesis detection problem as binary classification task rather than action localization. This is because emesis is a very infrequent behavior, and it lasts for a very short duration, typically one to two seconds. We find it hard to modify our model to fit the action localization protocol and to conduct a fair comparison of similar computational budget on the corresponding benchmark.

Therefore, we follow the reviewer's second instruction to conduct comparison. To be fair and considering our computational budget, we reduce the parameter number of ActionFormer to align with our model on training costs. As for training data, since the positive samples are relatively rare, we crop the video into clips with different starting time to generate multiple training data containing positive data points. We follow the action localization protocol to train ActionFormer. Accordingly, the action class number is changed to be 1 for emesis only. During testing, we treat it as positive as long as ActionFormer's prediction period overlaps with the ground truth, which is beneficial for ActionFormer. For both training and testing, we feed a sequence of 20 seconds to the model at once. As the model architectures are different, we tune the training step

number of ActionFormer and choose the model with best performance on validation set for comparison.

Table. Comparison of the accuracy of emesis detection in *Suncus murinus* between AED and ActionFormer

	Motion	RTX	Nicotine	Copper sulphate	Naloxone	U46619	Cyclophosphamide	Exendin-4	Cisplatin
AED	99.42%	100.0%	100%	100%	97.10%	98.97%	96.93%	98.91%	98.41%
ActionFormer	98.27%	97.10%	99.10%	100%	97.10%	95.88%	96.93%	95.65%	96.83%

As shown in the above table, AED outperforms ActionFormer on six out of nine metrics and achieves the same accuracy on other three metrics. We agree with the reviewer that emesis, as a type of infrequent and short action, may pose specific requirements on model design: emesis lasts for a very short period, which necessitates modules with local temporal bias like 3D convolution; considering emesis actions are relatively infrequent and independent, formulating emesis detection as action localization task (feeding a long sequence into model at once) may waste the model capacity as the temporal correlation among emesis actions are weak.

Minor points

(Line 166 and Fig.2) I felt that the word “batchfy” is confusing as “batch” is used in a different sense in artificial neural networks (as the authors did in line 172). In the vision transformer, it is termed patch partitioning and embedding.

Response: Thank you for the comments. We have replaced the term “batchfy” with “reshape” in both the text and figure to enhance clarity.

Revised text: page 7, line 169-170: “The input frames were transformed into feature maps using 3D convolutions, and then reshaped along the temporal axis.”

Revised figure 2.

(Line 246) The authors should solidify the quality of their dataset. It was unclear how skilled the human annotators were in the present format. The high score by the neural network can be achieved by systematically biased annotations. So, the accuracy of human annotation should be tested through inter-observer reliability.

Response: Thank you for the comments. The accuracy of human annotation has been cross-checked and verified by three experienced observers. We have included this statement in the revised manuscript.

Revised text: page 6, line 173-145: “The commencement and cessation times of each emetic episode were manually marked, cross-checked and verified by three experienced observers.”

(Line 444) Depending on the response to the previous comments, it seems that the main contribution of this paper is not conceptual advancement, but rather the implementation of the automated detection system for *Suncus emesis* and the development of its training dataset. If this is the case, the training dataset and the developed system should be made publicly available through cloud-based repositories. Ensuring easy access to these resources, not only through the manuscript but also by making the data and tools available, can significantly contribute to the research community. Ideally, the code should include a demo script for the training and evaluation to help users get started.

Response: We will share all the training and testing data, as well as the code, in a public repository that can be easily accessed by other researchers. We are happy to provide our tool to any researchers who need to automatically detect emetic events in this species. The links to the code and videos are shown in the data availability section.

Reviewer #3 (Remarks to the Author):

In this manuscript, Rudd et al developed a deep learning-based system for detecting emesis behaviors in *Suncus murinus* with high accuracy. Previous research analyzing emesis behaviors in *S. murinus* relies on manual scoring, which is laborious. Rudd et al demonstrates that their model can be used to efficiently and accurately score emesis behaviors in several conditions, including motion and with 8 different emetic drugs in short and long video durations. Together, this work addresses an unmet need in this field, and has great potential to be used by other researchers investigating emesis and drug screens.

I find this study well designed and conducted. The authors discussed most caveats and improvements that they could address in future works. For example, the accuracy decreases in longer videos, and the short emetic episodes can be missed. I have a few questions that just need to be clarified:

1. How reliable are the AED scoring? If you run the same testing video a few times, do you get the same results, especially for those long videos, or videos containing very short emetic episodes/body trembling?

Response: The AED scoring is highly reliable in the testing study. We obtain consistent results when we run the same testing video multiple times.

2. For the future work, if you train AED with all the videos including motion and 8 drug conditions, will it further improve the model to address the false negatives?

Response: Thank you for your comments. We believe that the model can be improved by incorporating additional training datasets into the existing ones. In our future work, we will continue to add more videos for training in order to develop a better tool for the scientific community to automatically detect emesis in *Suncus murinus*.

3. Will it work well if the video parameters are changed, for example, no bedding? How robust is the model to be used in a different lab video setting?

Response: Thank you for your comments. We have tested the performance of our tool when the observation box has no bedding inside, and the accuracy of the results is 100.0%. In our study, our aim is to establish a video recording system that is stable and reproducible. Our experimental design features a commonly used and straightforward video recording system for evaluating emesis behaviour in *Suncus murinus*, which can be easily replicated by other researchers. This will allow our tool to be widely shared within the scientific community.

Revised text: page 15, line 366-372: “Performance of the AED system in the motion-induced emesis model of *Suncus murinus* in the observation box without bedding

To assess the robustness of the AED system, we further evaluated its performance using a motion-induced emesis model in *Suncus murinus* in an observation box devoid of bedding. In this experimental setup, four male and female *Suncus murinus* were subjected to motion stimulation. We recorded an average of 9.7 ± 4.4 episodes of R + V, with the AED system demonstrating a prediction accuracy of 100% (Supplemental Figure S1).”

4. Deep learning-based systems for behavioral analysis has become more and more useful in the field of neuroscience. The authors also cited many key works (ref 11-15). In all cases, the training data and/or codes are publicly available for open access by other researchers for scrutiny, improvement and use. Is this what the authors are also considering? It will certainly be a great tool for the field.

Response: We will share all the training and testing data, as well as the code, in a public repository that can be easily accessed by other researchers. We are happy to provide our tool to anyone who needs to automatically detect emetic events in this species. The links to the code and videos are shown in the data availability section.

REVIEWERS' COMMENTS:

Reviewer #2 (Remarks to the Author):

I thank the authors for revising the manuscript. They have adequately addressed all of the concerns I raised in the previous review. I have only one remaining comment.

The authors have compared the accuracy of AED (proposed) with ActionFormer (an existing tool) for action recognition. The results demonstrated that their method outperformed ActionFormer on six out of nine tests and was comparable to the rest. However, the details of these results and the corresponding methods and discussion are missing from both the main text and supplemental materials. So, I recommend that these details be included in the paper to provide a more convincing evaluation of the proposed method.

Response: Thank you for the comments. We have included the details of the comparison results and the corresponding methods and discussion as suggested in the revised manuscript.

Revised text: page 11, line 249-255: “Comparison of the accuracy of emesis detection in *Suncus murinus* between AED and ActionFormer

As shown in Supplementary Table 2, the accuracy of emesis detection using ActionFormer in motion-, RTX-, nitcone-, copper sulphate-, naloxone-, U46619, cyclophosphamide-, exendin-4- and cisplatin-induced emesis model was 98.27%, 97.10%, 99.10%, 100%, 97.10%, 95.88%, 96.93%, 95.65% and 96.83%, respectively. AED outperforms ActionFormer on six out of nine metrics and achieves the same accuracy on other three metrics.”

Page 15, line 343-354: “In our study, we also compared the accuracy of emesis detection in *Suncus murinus* between AED and ActionFormer. ActionFormer is a localization model designed to detect and classify actions in videos by leveraging temporal attention and representation learning 38. It excels in tasks requiring fine-grained temporal segmentation, making it well-suited for scenarios with continuous or overlapping activities. The comparison between AED and

ActionFormer highlights the strengths of the developed model in detecting emesis in *Suncus murinus*. By framing the problem as a binary classification task instead of action localization, AED achieves remarkable accuracy across various stimuli, consistently outperforming or matching the performance of ActionFormer. Notably, AED's design effectively addresses the unique challenges of emesis detection, such as its infrequent occurrence and short duration, without requiring extensive modifications or additional computational resources.”

Page 20, line 468-480: “Emesis detection in *Suncus murinus* using ActionFormer
To ensure methodological fairness and accommodate our computational budget, the number of parameters was reduced in ActionFormer to align its training costs with those of our model. Regarding the training data, due to the relative scarcity of positive samples, the videos were segmented into clips with varying start times to generate multiple training instances containing positive data points. We adhered to the action localization protocol for training ActionFormer³⁸, subsequently modifying the number of action classes to a single category-emesis. During testing, a prediction is deemed positive if ActionFormer’s predicted temporal window overlaps with the ground truth, which advantages ActionFormer. For both training and testing phases, we input a 20-second sequence into the model at once. Given the differences in model architectures, we adjusted the number of training steps for ActionFormer and selected the model that demonstrated the best performance on the validation set for comparative analysis.”

Reviewer #3 (Remarks to the Author):

This manuscript is a great contribution to the research community. The authors have addressed all my questions, and I have no more questions after the revision.

Response: We thank the reviewer for evaluating our work.

Reviewer #4 (Remarks to the Author):

Dr. Rudd and colleagues have detailed an automated method for detecting emesis in *Suncus murinus* using a 3D CNN, reporting its high performance based on widely accepted measures of the field (precision, accuracy, F1beta). The generalization across several models of emesis is commendable, as is the pharmacological validation with palonosetron/cisplatin co-administration. Given *Suncus murinus*' place as a premier model organism for studying emesis, and the publicly available data set and model, this tool promises to further the field by relieving the burden of manual emesis scoring.

As a replacement to reviewer 1, I find the revision of this paper to adequately address the points previously highlighted, and think the manuscript is in a condition worthy of publication.

Response: We thank the reviewer for evaluating our work.